# A dynamic basal complex modulates mammalian sperm movement

Sushil Khanal[1], Miguel Ricardo Leung[2,3], Abigail Royfman[1], Emily L. Fishman[1], Barbara Saltzman[4], Hermes Bloomfield-Gadêlha [5], Tzviya Zeev-Ben-Mordehai [2,3 ✉] & Tomer Avidor-Reiss [1,6 ✉]

Reproductive success depends on efficient sperm movement driven by axonemal dynein-mediated microtubule sliding. Models predict sliding at the base of the tail – the centriole – but such sliding has never been observed. Centrioles are ancient organelles with a conserved architecture; their rigidity is thought to restrict microtubule sliding. Here, we show that, in mammalian sperm, the atypical distal centriole (DC) and its surrounding atypical pericentriolar matrix form a dynamic basal complex (DBC) that facilitates a cascade of internal sliding deformations, coupling tail beating with asymmetric head kinking. During asymmetric tail beating, the DC's right side and its surroundings slide ~300 nm rostrally relative to the left side. The deformation throughout the DBC is transmitted to the head-tail junction; thus, the head tilts to the left, generating a kinking motion. These findings suggest that the DBC evolved as a dynamic linker coupling sperm head and tail into a single self-coordinated system.

[1] Department of Biological Sciences, University of Toledo, Toledo, OH, USA. [2] The Division of Structural Biology, Wellcome Centre for Human Genetics, The University of Oxford, Oxford, UK. [3] Cryo-Electron Microscopy, Bijvoet Center for Biomolecular Research, Utrecht University, Utrecht, The Netherlands. [4] School of Population Health, College of Health and Human Services, University of Toledo, Toledo, OH, USA. [5] Department of Engineering Mathematics and Bristol Robotics Laboratory, University of Bristol, Bristol, UK. [6] Department of Urology, College of Medicine and Life Sciences, University of Toledo, Toledo, OH, USA. ✉email: z.zeev@uu.nl; Tomer.AvidorReiss@utoledo.edu

Reproductive success depends on the ability of sperm to swim through the female reproductive tract while out-competing their rivals[1–3]. Sperm motility is powered by dynein-mediated microtubule sliding in the axoneme[4–6]. The precise mechanisms that determine the flagellar waveform are unknown; however, several models were proposed[7]. The basal sliding model for mammalian sperm tail beating predicts sliding at the base[8,9], the site of the sperm centrioles; however, such sliding has never been observed[8,10]. Centrioles (aka basal bodies) are evolutionarily ancient organelles with a conserved architecture[11–13]. They are composed of nine compound microtubules (usually triplets) arranged symmetrically into a cylinder or barrel. The triplet microtubules are connected by various accessory proteins, forming a rigid structure restricting microtubule sliding[4]. Therefore, it is unclear how basal sliding takes place.

The sperm cell consists of a head and a tail linked by a neck (aka connecting piece or head–tail coupling apparatus)[14–16]. The neck contains two centrioles, a proximal centriole (PC) closer to the nucleus and a distal centriole (DC) at the flagellum base (Fig. 1a)[17]. In most eukaryotes, including humans, bovines, and most other mammals, the PC has a cylindrical shape, similar to canonical centrioles[16,18,19]; however, the composition and structure are slightly remodeled, e.g., the triplets have different lengths[20]. In contrast, the DC, in mammals, is dramatically remodeled and is atypical both in terms of composition and structure[20,21]. The most notable function of the sperm centrioles is after fertilization. They recruit egg pericentriolar martial (PCM) and form the zygote's first two centrosomes, emanating a large microtubule aster that helps bring the sperm and egg pronuclei together[22–26]. Since, similar to the canonical PC, the atypical DC functions post-fertilization, the reason for its atypical structure remains unknown.

The PC and DC are embedded in a specialized mass of atypical pericentriolar material: the segmented columns (SCs) and the capitulum[16]. Distally, the SCs are continuous with outer dense fibers associated with the microtubule doublets of the axoneme. Rostrally, the capitulum connects to the nuclear basal plate, forming the implantation fossa at the head-tail junction (Fig. 1a). How this basal multi-component assembly supports sperm movement is unclear, but it is usually modeled as a rigid structure that anchors the tail firmly, like a clamp, to the head, with little compliance allowed by the SCs[10,27] (see Supplementary Figs. 6 and 9a).

Here, we show that the sperm centriole inner scaffold splits into right and left rods associated with the splayed doublet microtubules of the DC, increasing its compliance. We find that these DC rods are asymmetric, and they have an opposite asymmetry to other DC substructures, the bars located in the DC center. Unlike the bars that stay overall static during beating, the DC rods and microtubules slide coordinately during the left-biased tail beating. The DC movement is also coordinated with the movement of the PC, SCs, and sperm head. These findings suggest the sperm neck structures, the DC, PC, and SCs, form a dynamic basal complex (DBC) that transmits the tail's microtubule sliding to the head.

## Results
### The centriole inner scaffold splits into two rods in the atypical centriole, increasing its compliance. 
Two protein classes maintain the rigidity of canonical centrioles. First, A–C linkers connect each triplet microtubule's A-tubule with the neighboring triplet's C-tubule[11,28]. Second, a cylindrical inner scaffold interconnects all triplets[29]. This scaffold includes the proteins POC1B, CETN1, POC5, and the two microtubule-binding proteins FAM161A and

WDR90; mutating these proteins destabilizes centriolar structure[30,31].

In the spermatozoon, the DC consists of splayed apart doublets instead of triplets[20], suggesting increased DC compliance. Furthermore, the inner scaffold proteins POC1B, CETN1, and POC5 reorganize into two-rod structures found between loosely clamped microtubules[21]. Here, we show that the microtubule-binding proteins FAM161A and WDR90 colocalize with the luminal and rod protein CETN1, labeling both at the DC and PC in human, rabbit, and bovine sperm (Fig. 1b, c). Like the other inner scaffold proteins, they appear mostly as two distinct rods in the DC (Fig. 1d–f, Supplementary Fig. 1a–d) and are enriched in the DC compared to the PC in all three species (Fig. 1g). This common localization pattern suggests that FAM161A and WDR90 are conserved components of DC rods. These observations suggest that the proteins form the scaffold that stabilizes typical centrioles split into two rods in atypical centrioles. This splitting could be an evolutionary innovation for reducing centriole rigidity, thus facilitating basal sliding.

DC rods vary in size across the mammalian species studied (Fig. 1h–j). This difference suggests that the atypical centriole that appeared early in mammalian evolution[18] continued to evolve in mammals, creating structural and functional diversity. The atypical centriole is largest in bovine sperm, and the theoretical foundation of basal sliding was based on bovine sperm[10]; therefore, we performed the remainder of our studies with bovine sperm.

### The DC rods and bars have opposite asymmetry. 
To gain insight into how DC rod proteins are situated relative to other sperm structures, we oriented straight sperm images with the PC on the right side[32] (Fig. 1a). We found that the rods are laterally asymmetric in bovine spermatozoa (Fig. 2a–d, Supplementary Movies 1–4). 3D stochastic optical reconstruction microscopy (3D-STORM) imaging of FAM161, POC1B, and POC5 showed that the left rods are consistently longer and thicker than the right rods. FAM161A and POC1B labeled ~50% longer rods ($P < 0.0001$) than those labeled by POC5, suggesting different protein locations within the DC (Fig. 2e). To determine the rods' relationship to the splayed microtubules, we measured tubulin-staining width across the DC (Fig. 2d). We found that the DC microtubule bundle's width is similar at the caudal end and 10% wider at the rostral end than that of the FAM161A and POC1B rods ($P = 0.2$, and $P < 0.0001$, respectively). FAM161A can simultaneously bind microtubules and other rod proteins[33], suggesting close rod-microtubule association at the DC lateral sides, and that the rods can act as scaffolds during basal sliding. This asymmetry agrees with the asymmetric flagellar waves[34], as well as the axoneme's structural and functional asymmetry[32,35]. The left four axonemal microtubules (doublets 4–7) work against the right three axonemal microtubules (doublets 1, 2, and 9) and the two stationary axonemal microtubules (doublets 3 and 8), generating a stronger left torque. Asymmetry in the neck is also observed in other mammals, suggesting that rod-asymmetry may be a more general feature of mammalian sperm[8,16,36–38].

To define the structural organization of the sperm neck in detail, we imaged unfixed, unstained bovine spermatozoa with cryo-electron tomography (cryo-ET) (Fig. 2f). Sperm cell images were rotated to view the PC on the right side. We compared PC dimensions measured by cryo-ET and STORM and found them consistent with each other (Supplementary Fig. 2a). We observed two electron-dense bars at the center of the neck, as reported previously[20,32,39] (Fig. 2f). Our data shows that the bars are intimately associated with the DC's central pair microtubules

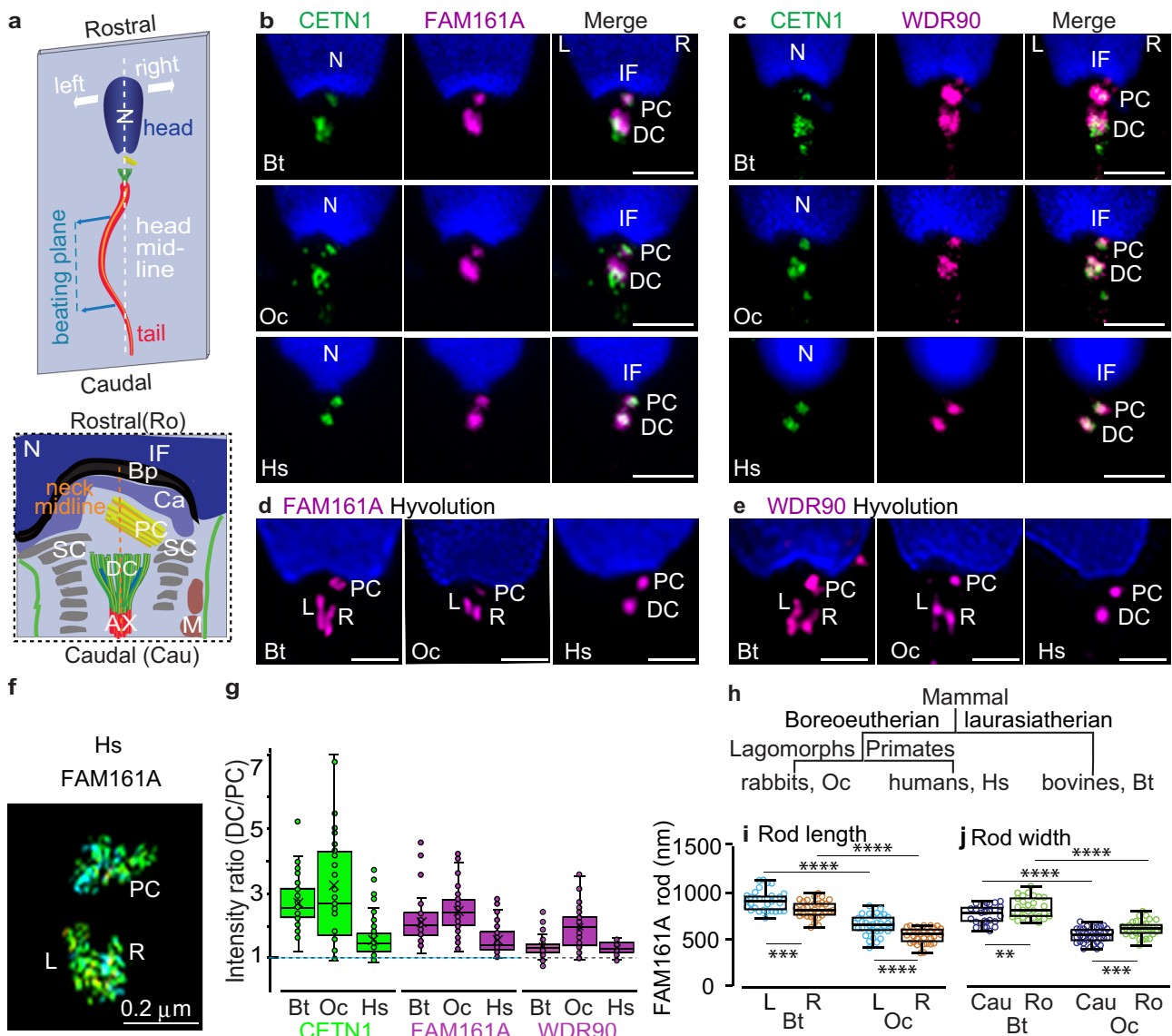

**Fig. 1 FAM161A and WDR90 are proteins in the atypical distal centriole. a** Schematics illustrating the sperm cell (top) and neck asymmetry (bottom). The head and neck asymmetry provide a basis for orientation in all figures, such that the top is rostral, the left side is on the left. Throughout the paper, images are oriented such that PC is at the right side of the connecting piece, and the distal tip of the PC points to the right side. Note that sperm beating occurs in one plane. **b**, **c** Confocal imaging with FAM161A (**b**) and WDR90 (**c**) staining. Both proteins partially co-localize with CETN1 at the PC and DC in sperm of bovines (*Bos taurus*, Bt), rabbits (*Oryctolagus cuniculus*, Oc), and humans (*Homo sapiens*, Hs). Scale bars 2 μm. **d**, **e** HyVolution imaging in bovine, rabbit, and human sperm with FAM161A (**d**) and WDR90 (**e**) staining. Scale bars 1 μm. **f** 3D-STORM imaging of human sperm with FAM161A staining. Note: STORM reflects labeling distribution, not intensity. **g** CETN1, FAM161A, and WDR90 intensity measurement from confocal images at DC and PC of bovine (n = 66 for CETN1 and FAM161A, n = 39 for WDR90), rabbit (n = 37 for CETN1, n = 80 for FAM161A, n = 46 for WDR90), and human (n = 62 for CETN1, n = 75 for FAM161A, n = 36 for WDR90) sperm. **h** Bovines (Laurasiatherian), humans (Boreoeutherian; Primates), and rabbits (Boreoeutherian; lagomorphs) represent three major phylogenetic groups of mammals. **i**, **j** FAM161A rod length (**i**) and width (**j**) measurement in bovines (n = 31 for length, n = 26 for width) and rabbits (n = 34 for length, n = 35 for width) based on confocal HyVolution imaging. Data are presented as box and whisker plots, where upper and lower bounds show interquartile range, line within the box shows median, and whiskers show minimum and maximum data points. Throughout the paper, the labels are the same: N nucleus, IF implantation fossa, PC proximal centriole, DC distal centriole, Bp basal plate, Ca capitulum, SC segmented column, Ax axoneme, M mitochondria, L left side, R right side, Cau caudal, Ro rostral. Statistical analysis, unpaired, two-tailed *t* test. **P < 0.01, ***P < 0.001, ****P < 0.0001; ns not significant. Source data are provided as Source Data File.

(Fig. 2f, g). The bars are asymmetric and are made of 1–4 plates of varying sizes separated by electron-lucent inter-plate material, revealing an unexpected level of complexity (Fig. 2g). Unlike the rods' "V" shape, the bars are nearly parallel to each other (Supplementary Fig. 2b). The caudal edge-to-edge gap between the smallest DC rods (POC5) is much larger (43%) than the corresponding gap between the bars (Supplementary Fig. 2c). The rods and bars have opposite asymmetry: the right bar has more

plates and is longer than the left (Supplementary Fig. 2d). The two bars are also situated closer to the DC's right microtubules (Fig. 2f and Supplementary Fig. 2d). We observe electron density associated with the lateral doublet microtubules' inner surface that corresponds to the rods' estimated location (Fig. 2f and Supplementary Fig. 2e). Altogether, these differences suggest that the central bars scaffold the central pair, while the rods scaffold the DC lateral side microtubules (Fig. 2e).

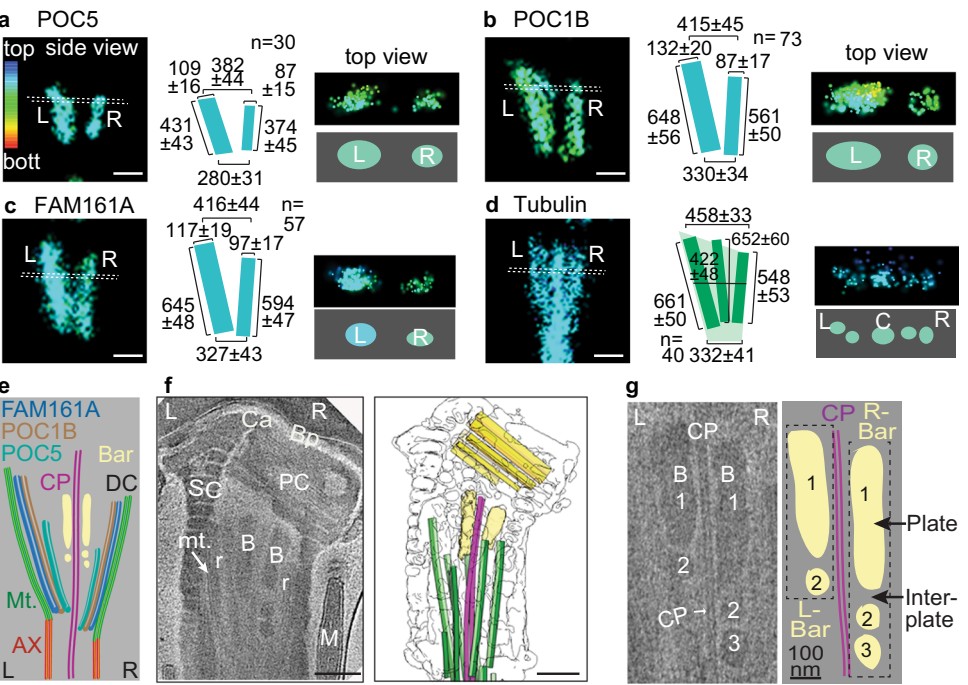

**Fig. 2 The DC is asymmetric. a–d** Representative 3D-STORM images taken with different rod protein staining. Rod proteins and microtubules are asymmetric in the bovine sperm DC, as observed using 3D-STORM. Side view (left two panels); top view (right two panels). All sizes are in mean ± sd nm. The colors represent depths of 1178 nm in the Z-plane, with red at the bottom (bott.) in all figures. Statistics are shown in Supplementary Fig. 2f. Scale bars 200 nm. STORM imaging with all proteins was repeated three times independently with a similar result. **e** Schematic diagram of a longitudinal section through the DC, showing relative positions of different structures based on 3D-STORM and cryo-ET imaging measurements. **f, g** A computational slice (left) and corresponding 3D segmentation (right) of a Volta phase plate cryo-tomogram of a bovine sperm neck (**f**). Digital zoom of a cryo-tomogram illustrating bar asymmetry, complexity, and intimate association with the central pair (**g**). The experiment was repeated twice independently with similar results. The labels "1", "2", and "3" mark distinct bar plates. Scale bars, 250 nm. C central microtubules, mt. microtubules, r rod, B bars, CP central pair. Source data are provided as Source Data File.

**DC rods and microtubules slide coordinately during the left-biased tail beating.** To gain insight into a possible reason for the intricate but atypical architecture of the DC, we analyzed sperm that were snap-frozen while actively swimming. We used the rod asymmetry as a reference to describe the flagellar waveform relative to the sperm head despite the cell's complex rolling motions[36]. We refer to this evaluation as centriole orientation-based sperm analysis (COSA). We classified the sperm images into four groups based on COSA, where the bigger rod is placed on the left and the PC on the right side of the head midline. We found that 15% had sharp left bends, 30% had mild left bends, 36% were straight, and 19% had slight right bends (Supplementary Fig. 3). Intriguingly, none of the sperm we analyzed ($n =$ 248) had a tail with a sharp right bend. The observation that all sharp bends were to the left and none were to the right indicates that the sperm tail has a bias toward the left side relative to the sperm head. A similar bias in the waveform was observed when staining for either FAM161A (Fig. 3a) or tubulin (Fig. 3b). The left-biased beat appears to have structural origins in the neck's inherent asymmetry, along with the increased rigidity imposed by the mitochondrial sheath that extends further rostrally on the right side of the neck (Fig. 1a)[40].

We then examined the DC substructures in chemically fixed sperm at distinct tail bending angles by 3D-STORM. In tails with sharp left bends, the right rod and microtubules are shifted rostrally relative to those on the left side (Fig. 3c, d, g, h). In contrast, the right rod and microtubules slide caudally relative to those on the left as the tail becomes straight or bent to the right (Fig. 3i, Supplementary Movie 5). We also observed that the DC central microtubule protrudes rostrally, which likely represents

the central pair (Fig. 3d). To examine the central microtubule's role during movement, we measured the distance from a centerline drawn through the PC's long axis to the left, center, and right of the DC microtubule bundle. As the tail bends from left to right, the DC's right microtubules move further away from the PC centerline, while the DC's left microtubules move closer (Supplementary Fig. 4). In contrast, the DC's central microtubules maintain the same distance from the centerline during the tail beating. Consistent with this, cryo-ET found that the central singlets are closer to the PC than are the DC's left and right microtubules and have the least change in distance from the PC centerline (Supplementary Fig. 5b). Also, the left and right bars associated with the central singlets were relatively static relative to each other during tail beating (Supplementary Fig. 5c–e). These differential movements suggest that the DC central pair and bars form a rail-like tracking system along which the rods and peripheral DC microtubules slide.

Our data provide direct evidence of microtubule sliding postulated by the basal sliding hypothesis[4], extending it to nanometer-scale shearing deformations in the neck. This hypothesis suggests that the tail's waveform is regulated by dynamic microtubule sliding at the axoneme base (Supplementary Fig. 6a, b). We tested the basal sliding hypothesis by comparing the sliding observed for DC microtubules and rods against three variables derived from the "sliding filament" hypothesis for flagellar movement[10,26,41]: the calculated average flagellum beating amplitude ($\bar{y}$), microtubule interfilament sliding along the tail ($\bar{\Delta}$), and the average waveform curvature ($\bar{\kappa}$). As expected from the observed beating asymmetry, all three waveform characteristics were skewed towards negative values,

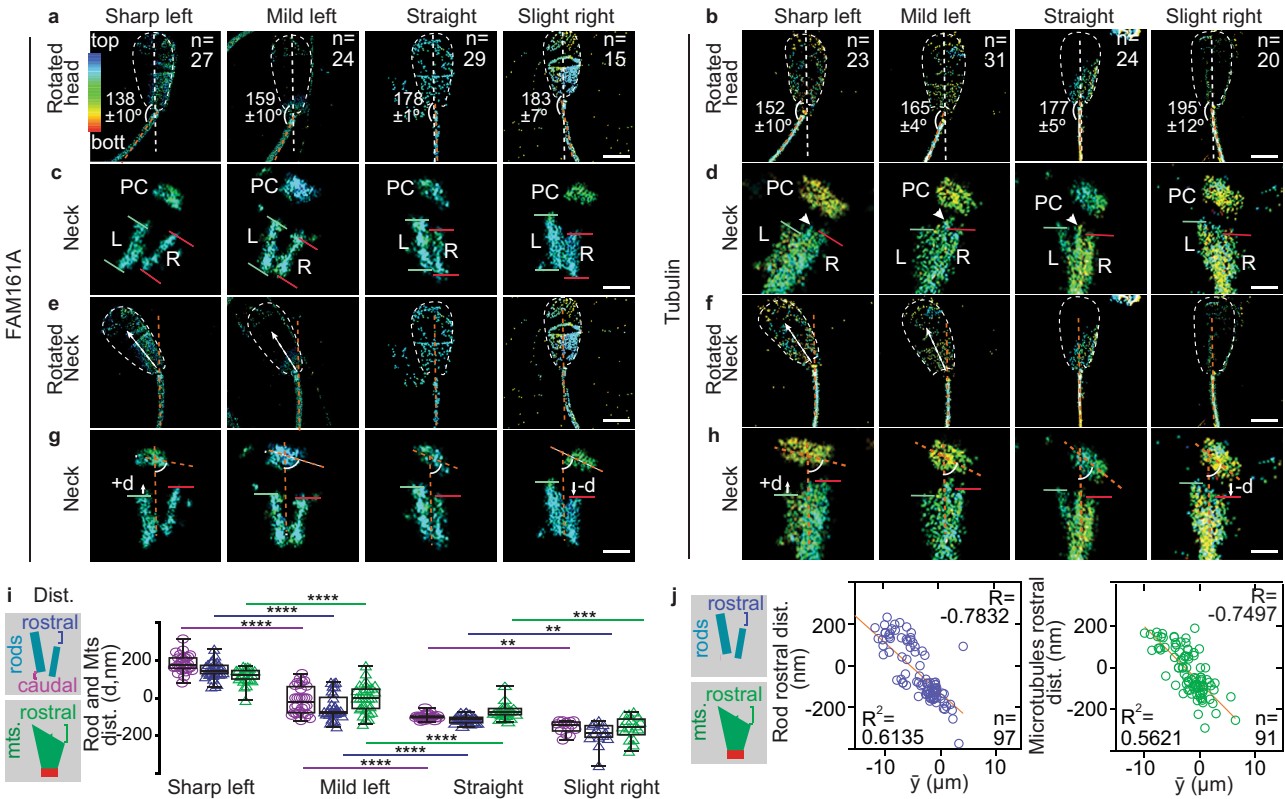

**Fig. 3 The DC's rods and lateral microtubules slide in correlation with tail beating. a, b** The four types of sperm tail bending oriented according to centriole orientation-based sperm analysis (COSA). Chemically fixed sperm were stained for either FAM161A (**a**) or tubulin (**b**). The white dotted line marks the head outline and midline, and the orange dotted line marks the neck midline. Head neck angles are shown as mean ± sd. **c, d** Zoom-in on the neck from panels **a** and **b** shows the location difference in the DC rod (**c**) and microtubule (**d**) edges. Arrowhead marks the central microtubules (**d**). Lines mark the left (L, green) and right (R, red) sides edges of each substructure. **e, f** STORM images rotated to orient the neck midline vertically. The white arrow shows how the head bends to the left during tail beating to the left. **g, h** Zoom-in on the neck from the panels (**e**) and (**f**). The orange dotted lines mark the midlines of the neck/DC. We define *d* as the distance between the rods and microtubules on the right side of the DC relative to those on the left side. Note that *d* is positive (+*d*) when the right side is higher than the left and negative (−*d*) when the right side is lower than the left. **i** Rostral (purple) and caudal (magenta) DC rod distance (dist.) and rostral DC microtubule distance (green) during the tail beating. The schematic on the left side of each graph represents the measurement scheme shown in the *y*-axis of the graph. Data are presented as box and whisker plots, where upper and lower bounds show interquartile range, line within the box shows median, and whiskers show minimum and maximum data points. All experiments were repeated three times independently with similar results. Statistical analyses used are an unpaired, two-tailed *t* test.****$p < 0.0001$, ***$p < 0.001$, **$p < 0.01$. **j** Correlation analysis between DC sliding at the rostral end (Y-axis) and tail beating amplitude ($\bar{y}$) (*x*-axis). The solid orange line in scatters plots represents the regression line. Every data point represents a cell. *R* Pearson correlation, $R^2$ linear regression. Correlation is statistically significant ($P < 0.0001$). Scale bars, 250 nm. Source data are provided as Source Data File.

the left side (Supplementary Fig. 7a, b). Basal microtubule and rod sliding show a strong correlation with the calculated averages of flagellar beating amplitude, sliding, and curvature (Fig. 3j, Supplementary Fig. 7c, d, Supplementary Fig. 8a, b). Overall, from sharp left to slightly right bent tail, the DC rod and the microtubules' basal ends are displaced 263–328 nm relative to each other. This sliding has the calculated order of magnitude of the flagellar control model fittings: 160 nm in bovine sperm[10] (Supplementary Fig. 8i). This similarity suggests that the DC sliding movement is related to the model predicted basal sliding; however, the observed sliding is more complex and includes associated structures such as the rods that are not accounted for in the sliding-filament model.

**A DBC transmits the tail's microtubule sliding to the head.** The current dogma holds that the neck is cemented to the head. Some movement was observed in the neck, but it lacked correlation with tail beating[32,42]. Therefore, the sperm head is thought to follow the tail's swimming movement passively[43,44]. In contrast, we observed a dramatic coordinated motion of neck structures

with a head movement, which we named "kinking" to signify a head movement relative to the sperm neck long axis (Supplementary Figs. 9 and 12).

We found elastic deformation in the neck beyond the DC. The right SCs are displaced relative to the left SCs during sperm movement (Fig. 4a, b). Also, the usually parallel segments of the SCs are bent between segments 8 and 9 in left curved sperm (Fig. 4c, d). The motion of the SCs causes the embedded PC to move, and the PC angle relative to the neck midline changes ~24° during tail beating (Fig. 4c, Supplementary Fig. 10a, Supplementary Movies 6 and 7). The PC also changes its lateral position relative to the neck midline, moving 140–200 nm to the left (Supplementary Fig. 10b). We found a high to strong correlation of PC position change with tail waveform variables, DC sliding, and SCs sliding (Fig. 5a–c, Supplementary Fig. 11a–d). Therefore, the neck deformation is due to coordinated displacements of DC, PC, and SCs during sperm tail beating.

Most significantly, we found a coordinated deformation inducing a head–neck kink, with the angle changing ~45°, causing the head to kink to the left when the tail beats to the left

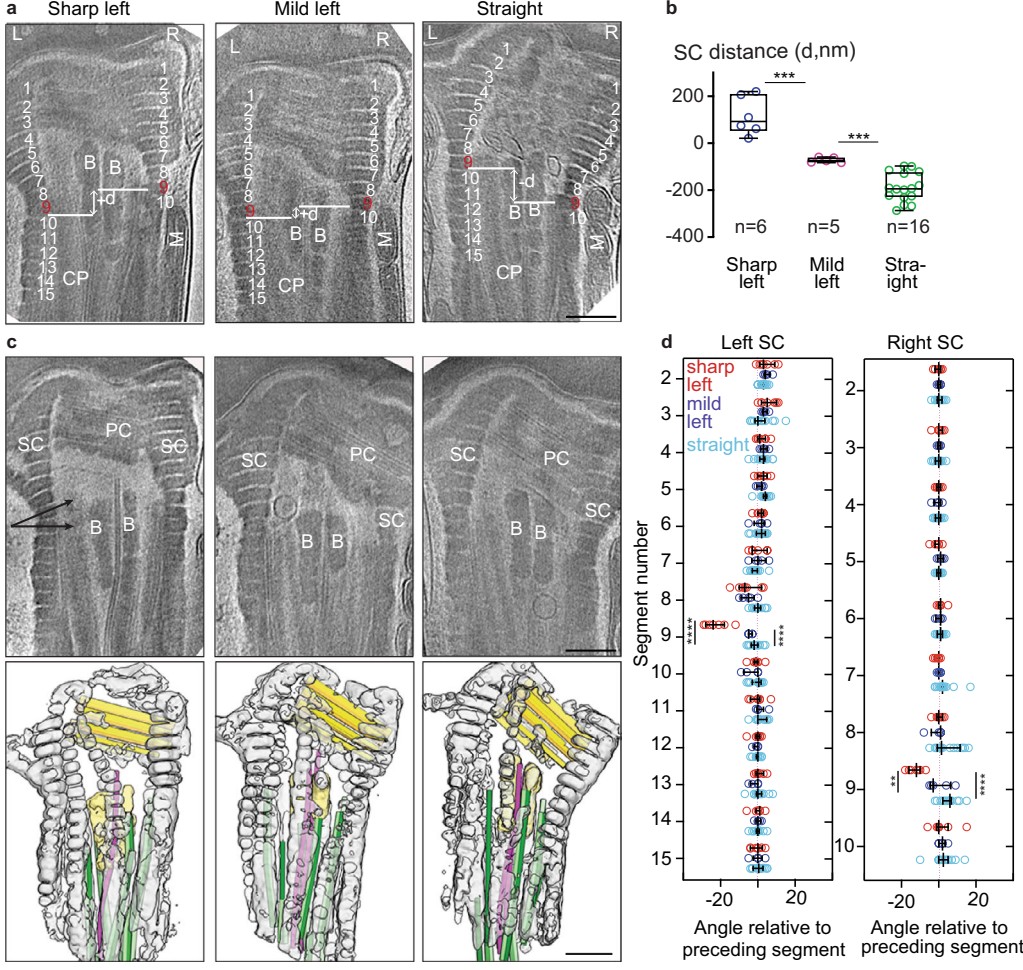

**Fig. 4 Segmented columns slide during sperm tail beating. a, b** Slices through Volta phase plate (VPP) cryo-tomograms of three types of sperm tail bending oriented according to COSA. Segmented columns (SCs) are labeled and numbered starting from the topmost segment as 1 and onward (**a**). Plot **b** shows the relative distance (**d**) between the left SCs segment and its corresponding right SCs segment. Negative values indicate that segment 9 on the right is lower than segment 9 on the left ("−d"), while positive values indicate that segment 9 on the right is higher than segment 9 on the left ("+d"). Data are presented as box and whisker plots, where upper and lower bounds show interquartile range, line within the box shows median, and whiskers show minimum and maximum data points. Experiment was repeated twice independently with similar results. **c** Computational slices through cryo-tomograms of sperm with varying extents of tail bending (upper panels) and corresponding three-dimensional segmentations (lower panels). Labels: SC segmented columns, B bars. Color scheme: gray: segmented columns, dark yellow: proximal centriole triplets, green: distal centriole doublets, pink: central singlets, pale yellow: bars. Two angular arrows in the upper panel mark the area where the SCs bend when the head kinks to the left. **d** Relative angles between each segment. Every data point represents a cell, and lines represent mean ± sd. The experiment was repeated twice independently with similar results. All statistical analyses used are an unpaired, two-tailed t-test. **P < 0.01, ****P < 0.0001. Source data are provided as Source Data File. Scale bars: 250 nm.

(Fig. 3e, f, Supplementary Fig. 11e, Supplementary Fig. 12, Supplementary Movie 8). A similar kink was observed in the past in live and reactivated bovine sperm[45,46], and it may contribute to the rapid wiggling of the sperm head around swimming averaged path[34,47]. Interestingly, we found that head kinking also correlates with tail variables and neck structure (DC, SCs, and PC) deformation during tail beating (Fig. 5c, Supplementary Fig. 11a–d, right panels), suggesting that the neck kinking is in coordination with other nanometric structural deformations of the neck during tail beating. This head kink is marked by a sharp angle between the tail's tangent angle at the neck, where both the head and tail bend to the left (Fig. 5d). The coordinated tail bending and head kinking suggest a dynamic structural modulation during swimming.

Sperm tail movement drives neck deformation and heads kinking, as axonemal dyneins are the only known active motor proteins during sperm swimming. Two mechanisms may translate forces from the tail: the axoneme's attachment to the

DC and the tail's outer dense fiber attachment to SCs. Significantly, exploratory factor analysis of three sets of data measuring a total of 21 variables during sperm beating indicated that three factors explained the underlying sperm behavior: a major tail-to-head coordinated movement, no movement of DC center and width, and a mixture of the two (Supplementary Fig. 13). These data support a model in which the axoneme sliding that generates tail beating also deforms the neck, subsequently kinks the head (Fig. 5e).

## Discussion

Centriole structure and function are conserved across ciliated cells, from protists to mammals. Therefore, it is surprising that the centriole found at the flagellum base is structurally atypical in mammalian sperm[48–51]. However, the atypical centriole is the ideal mechanism to allow basal sliding. The basal sliding hypothesis was postulated before discovering the atypical centriole and assumed that the axoneme base moves relative to a

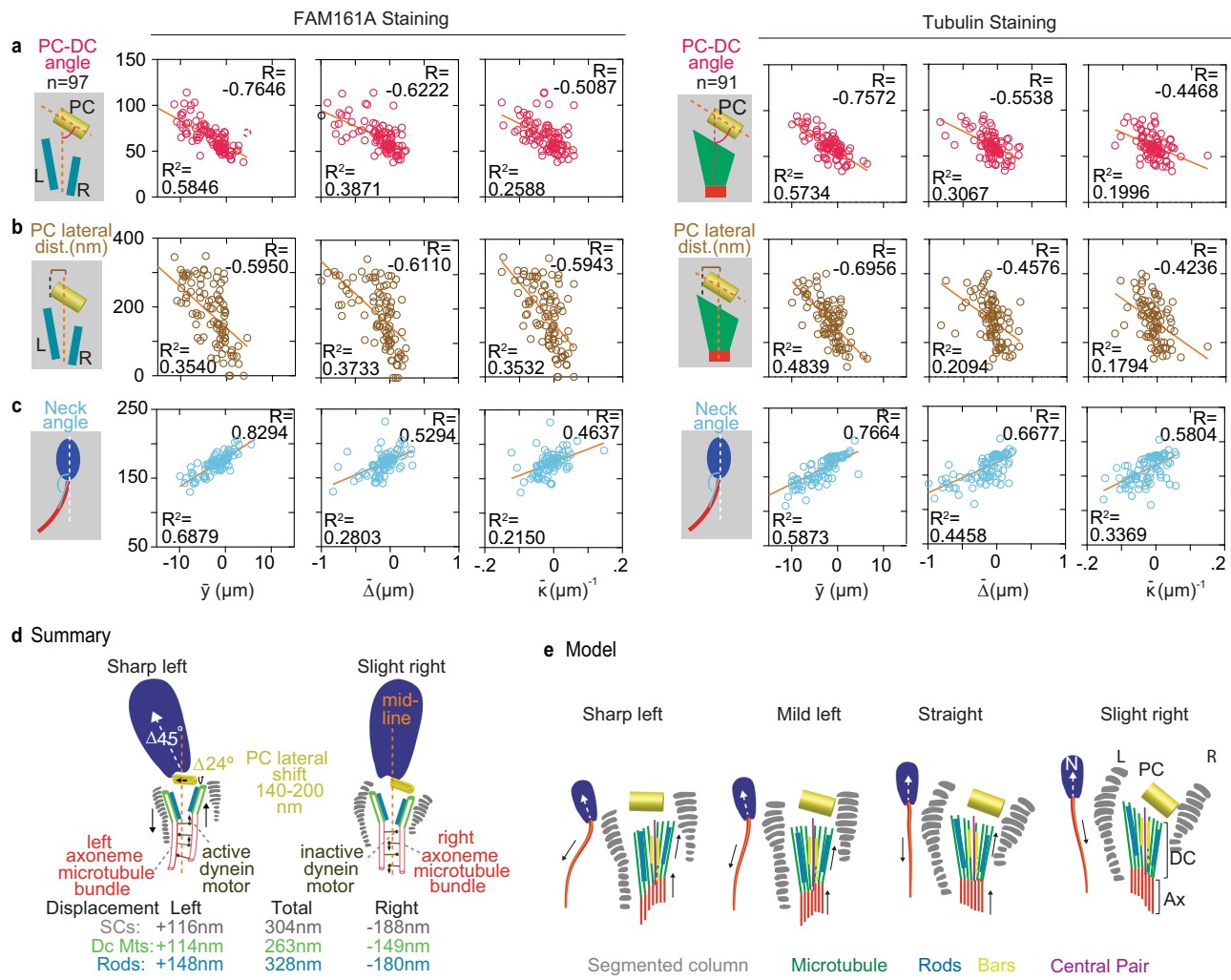

**Fig. 5 Sperm tail beating correlates with PC and head position. a–c** Correlation analysis between tail variables, beating amplitude ($\bar{y}$), interfilament sliding ($\bar{\Delta}$), curvature ($\bar{\kappa}$), and PC–DC angle (**a**), PC lateral shift (**b**), and head–neck angle (**c**) with FAM161A (left panels) and tubulin (right panels) staining. The schematics shown to the left of each graph represent the measurements plotted on the y-axis. All the tail variables are plotted on the x-axis. $R$ Pearson correlation, $R^2$ linear regression. Each circle represents the data point for an individual cell. All experiments were repeated three times independently with similar results. All Correlation analyses in **a–c** are significant ($P < 0.0001$). **d** The summary of overall magnitude in deformation of DC, PC, SCs, and head kinking between sharp left beating cells to slight right beating cells. **e** Model showing deformation of DC, PC, SCs, and head kinking during sperm tail beating from sharp left to slight right. Source data are provided as Source Data File.

static basal complex (i.e., the SCs) and relative to the other axonemal microtubules (Supplementary Fig. 6a, b). This sliding movement was estimated to be 160 nm in bovine sperm using the basal sliding model[10]. Here, we show that the DC rods attached to the DC doublets move relative to each other and relative to central microtubules and their associated bars (Fig. 3c). The movement we observed is about twice the predicted one, and this difference may be explained by the unexpected movement in the basal complex in the SCs, the PC, and the sperm head. We, therefore, refer to the sliding element at the base of the axoneme as the DBC. We postulate that the DBC has two functions: (i) it shapes tail beating according to the basal sliding hypothesis; and (ii) it translates the axoneme's piston-like tangential movement into a cascade of multi-component shearing deformations culminating in a coordinated head kinking motion.

A potential advantage of mechanically coupling tail curvature with the head attachment angle is that the head can impact tail movement to provide mechanosensory information, thus providing a way for the sperm to better navigate the various female reproductive tract barriers. As such, we hypothesize that the DBC

may act as a morphological computer[52], regulating tail beating from external feedback imparted to the head during sperm navigation. Therefore, the DBC may have evolved to serve as a mechanotransducer, coupling sperm head and tail into a single self-coordinated system. This coupling may be advantageous in internal fertilizers such as mammals during sperm interaction with background flows, as in rheotaxis, obstacles, and boundary following navigation near the wall of the female reproductive tract[53,54]. This coupling may also help the sperm dig its way through the external protective shields surrounding the ovum via bending modulation. We show that in bovine sperm, this coupling associate with asymmetrical tail beating and head kinking to the left and may help achieve forward swimming movement via sperm rolling motion[36]. However, small changes to neck components will likely result in distinct movement patterns, assisting in creating a spectrum of sperm behaviors in other animal species. Altogether, the DC's properties suggest that it evolved by repurposing centriolar proteins to assemble a transmission system (the DBC) that couples the flagellar motors to the whole sperm, thereby enhancing sperm function.

## Methods

**Sperm sample**. Ejaculated spermatozoa were donated by Dr. Bo Harstine at Select Sires Inc. (bovine, Holstein) and Dr. Jie Xu at the University of Michigan (New Zealand rabbit) and purchased from Manhattan Cryobank and Fairfax Cryobank (human). Human sperm samples were acquired with approval from the University of Toledo's Institutional Review Board (IRB).

**Preparation of sperm for studying asymmetric beating and frequency distribution**. We used the swim-up technique to select motile bovine sperm for snap freezing. The swim-up technique was performed using the PureSperm Wash kit (Nidacon, PSW-100) as instructed. Briefly, one straw of cryopreserved bovine semen (0.5 mL; $40 \times 10^6$ sperm/mL) was thawed and placed in a round-bottom 15-mL tube. 1 mL of PureSperm wash solution (PSW-100) was placed on top of the semen. The tube was slanted at 45° and incubated at 37 °C for 1 h. After incubation, the upper layer of approximately 0.75 mL was removed and placed in a new tube. Motility was assessed under a microscope, and samples with more than 90% motile sperm were used for further processing. The collected upper layer was then centrifuged at 250$g$ for 20 min. The pellet was suspended in a wash buffer (PSW-100) and centrifuged at 250$g$ for 8 min. The pellet was resuspended in a minimal volume of wash buffer to concentrate the sperm. Approximately, 15–20 µl of the sample was placed on a glass slide, covered by cover glass, and immediately dropped into liquid nitrogen for snap freezing. A snap-frozen sample was processed for immunostaining as described in the Immunofluorescence section.

**Study of asymmetric beating and occurrence distribution**. Snap-frozen sperm were stained to locate the PC and DC within the cells. Images were captured by randomly selecting several fields of view. All cell images were rotated to orient the head straight, and a head midline was drawn. The straight head sperm cells were oriented to place the implantation fossa on the left side, and the PC on the right of the head midline. After that, the sperm cells were characterized as *sharp left, mild left, straight*, and *slight right*, based on tail bending direction relative to the head midline. Mild left cells displayed a mild kink at the neck or mild bending at the mid-piece to the left of the midline. Cells were characterized as straight when the head, mid-piece, and most of the principal pieces were in line with the head midline. Similarly, slight right bent cells displayed a mild bending of the tail to the right of the head midline with a mild kink at the neck or mild bending at the midpiece. None of the cells displayed a sharp kink at the neck when tail beating occurred to the right. The sharp left cells were further characterized as Type 1 when there was a very sharp kink at the neck and Type 2 when there was a kink at the neck and a strong bend at the mid-piece. Similarly, we characterized mild left cells as Type 1 and Type 2, based on whether they showed a kink at the neck or bending at the mid-piece.

**Preparation of sperm for study by immunofluorescence**. Motile ejaculated spermatozoa were selected by 40/80 density gradient using a PureSperm Wash kit (Nidacon, PS40-100, and PS80-100). A sperm pellet sample with at least 70–80% motility was used for further processing. The sperm pellet was resuspended in wash buffer (PSW-100). Approximately, 7 µL of cleaned spermatozoa were placed on a glass slide, and a Sigmacote coverslip (Sigma, SL2) was placed on top. The entire slide was then snap-frozen and stored in liquid nitrogen. Sperm slides were stored in liquid nitrogen until staining.

**Preparation of sperm for stochastic optical reconstruction microscopy (STORM)**. For STORM, motile sperm were selected by either density gradient or the swim-up technique. The swim-up technique for STORM was performed in two different buffers, wash buffer (PSW-100) or modified Krebs–Ringer Hepes (mKRH) medium (pH 7.4, lacking $CaCl_2$, $NaHCO_3$, and Bovine Serum Albumin) composed of 94.0 mM NaCl, 1.19 mM $MgSO_4 \cdot 7H_2O$, 1.19 mM $KH_2PO_4$, 4.78 mM KCl, 25.07 mM HEPES, 27.64 mM glucose, 50 mg/ml streptomycin sulfate and 100 IU/ml penicillin G potassium salt. Motile sperm were placed on round cover glass (Fisher Scientific, 72231-01) or MatTek glass-bottom dish (MatTek corporation, p35G-1.5-10.C), allowed 5 min for movement, and snap-frozen in liquid nitrogen. The snap-frozen sperm were fixed in prechilled methanol for 3 min and prepared for immunostaining as described in the "Immunofluorescence" section.

**Preparation of sperm for cryo-electron tomography (cryo-ET)**. For cryo-ET, bovine semen was collected from Holstein bulls (CRV Delta, The Netherlands) and prepared for imaging within hours of collection. Semen was diluted to a sperm concentration of $\sim 30 \times 10^6$ cells/mL with either Tris-citrate buffer (96 mM Tris, 27 mM fructose, 35 mM sodium citrate, pH 7, 300 mOsm) or OptiXcell buffer (IMV technologies, 026218–025239). Sperm motility was assessed with a computer-assisted sperm analysis (CASA) system and found to be ~83% and ~85% for sperm diluted in Tris-citrate and OptiXcell, respectively. Sperm suspensions were further diluted to a concentration of $\sim 3 \times 10^6$ cells/mL, and approximately 3 µL was pipetted onto glow-discharged Quantifoil R 2/1 200-mesh holey carbon grids. One microlitre of a suspension of bovine serum albumin (BSA)-conjugated 10-nm gold beads (Aurion, 210.033) was added, and the grids were then blotted manually from the back (opposite the side of cell deposition) for ~3 s. Grids were immediately plunged into liquid ethane cooled to liquid nitrogen temperature, and stored under liquid nitrogen until imaging.

**Antibodies**. Primary and secondary antibodies were purchased from various suppliers (Supplementary Table 1). Concentrations of various antibodies used for confocal and STORM microscopy are listed in the table below.

**Immunofluorescence**. For staining, slides were removed from liquid nitrogen, the coverslip was removed using forceps, and the slide was placed in a pre-chilled Coplin jar of ice-cold methanol for 3 min. Next, the slide was placed in phosphate-buffered saline (PBS) for 1 min and then set for 60 min in fresh PBS with 0.3% Triton X-1000 (PBST) at room temperature for permeabilization. Then blocking was performed in PBST-B. PBST-B was prepared by adding 1% bovine serum albumin (BSA) to PBST, and slides were then placed in PBST-B for 30 min. Primary antibodies diluted in PBST-B were added to the slides, after which the slides were covered in parafilm, set in a humidity chamber, and incubated overnight at 4 °C. The slides were washed three times in PBST for 5 min each. Next, the secondary antibody mixture was prepared by adding secondary antibody and Hoechst (#H3569, Thermo-fisher Scientific, 1:200) to PBST-B solution. The secondary antibody mixture was added to the slides, which were then covered in parafilm and incubated for 2 h at room temperature. Slides were then washed three times with PBST for 5 min each, followed by three washes with PBS for 5 min each. The slides were covered by cover glass (18 × 18 mm), sealed by clear nail police, and were stored at 4 °C until imaged.

For STORM, immunostaining was performed as described above with slight modifications. Incubation in primary antibody was done for 24 h at 4 °C, and incubation in secondary antibody was done for 4 h at room temperature. Washes after primary and secondary antibody incubations were completed five times for 5 min with each solution. After washes, the cover glass or MatTek dishes were stored in PBS at 4 °C. Imaging was performed within 24–48 h.

**Confocal imaging**. Slides were imaged using a Leica Sp8 confocal microscope, and some images were processed using a Leica HyVolution 2 System. Images of sperm were captured at a magnification of 63× and a zoom of 6×, with 512 × 512 pixel density. Using Photoshop, immunofluorescence sperm images were cropped to 200 × 200 pixels or 65 × 65 pixels. The images' overall intensity was modified to allow easy visualization, and panels were resized to 1 × 1 inch. and 300 DPI for publication.

**3D-STORM Imaging**. 3D STORM imaging was performed on a Nikon N-STORM4.0 system using an Eclipse Ti inverted microscope, an Apo TIRF 100× SA NA 1.49 Plan Apo oil objective, 405-, 561-, 488-, and 647-nm excitation laser line (Agilent), and a back-illuminated EMCCD camera (Andor, DU897). The 647-nm laser line was used to promote fluorophore blinking. A 405-nm laser was used to reactivate fluorophores. ~30,000-time points were acquired at a 20 Hz frame rate, each 16–20 ms in duration. NIS-Elements (Nikon) was used to analyze and present the data. For imaging samples on cover glass, the cover glasses were mounted on a depression slide in imaging buffer (10% dextrose in 100 mM Tris at pH 8.0, 25 mM β-Mercapto-ethylamine, 0.5 mg/mL glucose oxidase, and 67 µg/mL catalase). The cover glass was sealed with Body Double SLK (SO56440A and SO5644B) and allowed 3 min to air dry, after which the sample was processed for imaging. For imaging samples on a MatTek Dish, 1 mL of imaging buffer was placed into the dish and imaged under the STORM microscope. During imaging, different types of cells (i.e., sharp left, mild left, straight, and right bent) were selected based on the criteria explained above in the section "study of asymmetric beating and occurrence distribution." All STORM imaging for each figure was replicated at least three times with the first bull's sperm and at least once with two other bulls' sperm. Data shown in all figures are cumulative of all replicates.

Z-Calibration was done using florescent beads and stored as a file for each specific objective and buffer condition according to the manufacturer's instructions. In our STORM, images were taken using 100X objective and the same buffer (see above). The raw data obtained by image acquisition for all STORM images were analyzed using NIS-Elements. Analyzed images were exported in TIF format. Publication-ready STORM images were prepared in Photoshop by cropping to 300 × 300, 100 × 100, or 65 × 65 pixel sizes and resizing cropped images to 1 × 1 inch and 300 DPI. The background intensity of the whole image was enhanced to clearly visualize the sperm head.

**Cryo-electron tomography**. Tilt series acquisition—tilt series were acquired on a Talos Arctica (ThermoFisher) operated at 200 kV. The microscope was equipped with a post-column energy filter (Gatan) in zero-loss imaging mode with a 20-eV energy-selecting slit. All images were recorded on a K2 Summit direct electron detector (Gatan) in counting mode. Tilt series were collected using SerialEM (Mastronarde, 2005), with a Volta phase plate (VPP)[55], and at a target defocus of −0.75 µm. Tilt series were typically acquired in 2° increments over a range of ±50° using a grouped dose-symmetric tilt scheme with groups of 3 tilts.

Tomogram reconstruction, segmentation, and analysis—frames were aligned on the fly using Warp[56]. Tomograms were reconstructed in IMOD[57] using weighted back-projection with a SIRT-like filter[58] applied for visualization and

segmentation. VPP tomograms were not CTF-corrected. Segmentation of the connecting piece was first performed semi-automatically using the neural network-based workflow implemented in the TomoSeg package in EMAN 2.21[59] and manually refined in Chimera 1.12[60]. Microtubules, however, were traced manually in IMOD. Visualization was performed in Chimera 1.12. Measurements were performed either in Fiji or in IMOD on central ~20-nm-thick slices through tomograms filtered with 20 iterations of a SIRT-like filter.

**Intensity measurement by photon counting**. For intensity measurements of the various proteins, images were captured using a confocal microscope in counting mode. All images were captured using a constant laser power of 5%. The images were then analyzed for protein intensity using the Leica LASx program. Briefly, a round region of interest (ROI) 1.5 μm in diameter was drawn to encompass the PC and DC, and pixel sum intensity was recorded for each ROI and exported into a Microsoft Excel sheet for additional calculations.

**STORM image quantification**. PC and DC dimension measurements: STORM image measurements were performed using Nikon's NIS-Elements imaging software. The accuracy of measurements in NIS-Elements was confirmed by measuring the diameter of the tubulin-stained sperm axoneme. Additionally, the determinations of tubulin-stained PC's dimensions by STORM and Cryo-ET were comparable. No STORM images were excluded for quantification. All the images were quantified for all parameters as much as possible. Some of the rod parameters were not measured in some images because rods were not distinct enough to separate.

To measure rod length and width, a line, starting at 50% of the first intensity peak through 50% of the last intensity peak, was drawn and measured along the rods' length and width.

Caudal and rostral distance measurements: Two distances (caudal and rostral) were measured for rod sliding analysis. Since the DC microtubules are connected to the axoneme, only their rostral distance was measured. Sperm cells were first rotated to make neck straight and were oriented to maintain the PC tip pointing to the right side and bigger rod on left side of neck midline. This consistent reference was used for defining the left-right rod and microtubules. Measurements were taken at the caudal and rostral ends of the right-side rod and microtubules relative to the left-side rod and microtubules at the respective side. A measurement was assigned a "−ve" value when the right rod or microtubule was below the left rod or microtubules and a "+ve" value when the right side was above the left side.

**Waveform amplitude $\bar{y}$, sliding $\bar{\Delta}$, and curvature $\bar{\kappa}$ of the sperm flagellum**. We calculated three variables derived from the flagellum waveform to infer the "sliding filament" hypothesis during tail beating[10,27,35,61–63]: average flagellum beating amplitude ($\bar{y}$), microtubule sliding along the tail ($\bar{\Delta}$), and average waveform curvature ($\bar{\kappa}$). Importantly, the flagellum amplitude was measured relative to the sperm head orientation; however, curvature and microtubule sliding are quantities independent of head orientation.

The flagellar waveform was extracted with a semi-automated bespoke image processing algorithm in MATLAB to extract the coordinate values of every point along the flagellum relative to the sperm head $r(s) = (x(s), y(s))$, parametrized by arclength $s^{27}$. All flagellar shapes are rotated and translated so that the long axis of the sperm head is aligned with the x-axis, while the sperm neck is centered at the origin. The deflection of the tail shape in the y-direction thus captures the waveform's amplitude, with positive (negative) values for flagellar points lying on the right (left) in respect to the head. The right and left sides of the sperm head are determined by the proximal centriole's conserved position, chosen here to always lean toward the head's right side. We refer to this as COSA, as detailed in the Main Text. Average flagellar deviation from the x-axis along arc length is denoted by $\bar{y}$ and captures the average waveform deflection or amplitude relative to head orientation.

We consider a sliding filament model of the sperm flagellum for estimation of flagellar interfilament sliding[63]. The sliding filament model abstracts the flagellum using a two-dimensional representation composed of two filaments. Each constituent filament $r(s) = r(s)\frac{a}{2}\hat{n}(s)$ is separated by a distance $a$ (flagellar diameter) normal to the flagellum centerline $r(s)$ at every point in arclength $s^{10,27,35,61–63}$. Geometry constrains the normal vector $\hat{n}(s) = (-\sin\theta, \cos\theta)$ to the plane, where $\theta(s)$ is the angle between the fixed-frame x-axis and the tangent to the centerline $\hat{n}(s) = r_s$, where subscripts denote derivative in respect to s. Like a rail-track, the constituent filaments travel distinct contour lengths, forcing a geometrical arclength mismatch $\Delta_T - \Delta_0 = \Delta(s) = a(\theta(s) - \theta_0)$, where $\Delta_0$ and $\theta_0$ are the length mismatch and tangent angle, respectively, at $s = 0$. $\Delta_T$ is the total interfilament sliding along the flagellum, whilst $\Delta(s)$ captures the incongruence of interfilament sliding caused solely by waveform curvature, and thus in the absence of basal sliding $\Delta_0$. This is referred to here as flagellar sliding $\Delta(s)$. From this, $\bar{\Delta}$ captures the average flagellar sliding along arclength. For the calculation of $\Delta(s)$, we took the flagellar diameter $a = 600$ nm. The signed waveform curvature is simply $\kappa(s) = \theta_s$, and $\bar{\kappa}$ denotes the average curvature along arclength.

**Exploratory factor analysis (EFA)**. The statistical grouping technique known as exploratory factor analysis (EFA) was utilized. First, s for consideration in the EFA was checked for normality. EFA combines variables into clusters based on their

collinearity. These groupings can be further utilized to reduce the number of variables used as predictors in multivariable analyses. Highly correlated variables are combined to describe a factor. The 'loading' of these factors can be visualized as the Pearson correlation between each variable and its factor. The resulting factors are not collinear with each other and thus can be used together in multiple regressions. The varimax orthogonal rotation method was utilized. The Kaiser–Meyer–Olkin measure of sampling adequacy was calculated (>0.5 indicates sample adequacy), and Bartlett's test of sphericity was used to assess matrix correlation between the variables using an alpha of 0.05 to indicate significant correlation. A scree plot was used to assess the number of factors in the model. EFA was carried out with a varimax rotation method. Variables were considered to significantly contribute to their factor if their factor loadings were greater than 0.40. The EFA was completed using SPSS v2 (IBM Corp, Version 26.0., Armonk, NY).

**Statistical analysis and reproducibility**. Unless otherwise noted, each experiment was performed at least three times independently with similar results. All averages and standard deviations in this study were calculated in a Microsoft Excel Sheet. All correlations, regressions, and t test analyses were performed using GraphPad Prism 8.0. The number of cells analyzed (n) and all P values are stated in each figure or figure legend. The statistical analysis performed in this study is an unpaired, two-tailed t test. All quantification data are presented as Box and Whisker and scatter plots. All box and whisker plots are represented as a minimum to maximum, showing all data points with medians and interquartile range. Each data point in all scatter plots represents a measurement for an individual cell. P values are indicated as asterisks (*) highlighting the significance of comparison: $*P < 0.05$, $**P < 0.01$, $***P < 0.001$, $****P < 0.0001$.

**Reporting summary**. Further information on research design is available in the Nature Research Reporting Summary linked to this article.

## Data availability
Relevant data supporting the findings in this study are available in this paper and supporting information file, and from the corresponding authors upon reasonable request. Source data are provided with this paper.

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

## Acknowledgements

We would like to thank Kebron Assefa, Mohamed Baker Nawras, and Katriana Turner for their assistance throughout the study. We would like to thank The University of Toledo Instrumentation center for access to STORM microscopy. We thank the following people for providing sperm: Dr. Bo Harstine from Select Sires Inc. for donating the bovine sperm, Dr. Jie Xu lab at the University of Michigan for rabbit sperm, and Dr. Steve Pool at Fairfax Cryobank for human sperm, Dr. Heiko Henning at Utrecht University for providing bovine sperm for Cryo-ET. We thank Dr. Mihajlo Vanevic for cryo-EM computational support, Dr. Stuart Howes, Ingr Chris Schneijdenberg, and Johannes Meeldijk for management and maintenance of the Utrecht University EM Square facility. We thank the following people for editing and advising: Dr. Bo Harstine and Dr. Jadranka Loncarek. This work was supported by grant number R21 HD092700 from Eunice Kennedy Shriver National Institute of Child Health and Human Development and a grant from Select Sires Inc. This work was funded by NWO Start-Up Grant 740.018.007 to T.Z., and M.R.L. is supported by a Clarendon Fund-Nuffield Department of Medicine Prize Studentship.

## Author contributions

T.A.R. and S.K. conceived and presented the idea. S.K. carried out the confocal microscopy and STORM. M.R.L. and T.Z.B.M. carried out cryo-ET. H.B.G. carried out mathematical and waveform analysis. A.R. assisted in STORM. E.L.F. assisted in confocal microscopy. B.S. carried out the multiparametric analysis. All authors provided critical feedback and helped to shape the research, analysis, and writing the paper.

## Competing interests

The authors declare no competing interests.
