## [Peer Review File · Nature Communications]

Reviewer #1 (Remarks to the Author):

In this study, Khanal and colleagues use super-resolution microscopy (SRM) and volta phase plate cryo-tomography (cryoET) to demonstrate that the region between the head and the tail of the sperm cell (head-tail junction) is deformed upon tail beating and head kinking. It is a very beautiful analysis that combines state-of-the-art imaging techniques and demonstrates for the first time that distal centriole (DC) deforms under the effect of flagellar beating and that the surrounding protein complexes also undergo deformation. These observations provide a better understanding of what happens at the nanoscale during the flagellar beating. I think that this article is of high importance, nevertheless, I think that some points need to be clarified before publication:

- Initially, the authors localize several proteins at the DC level. They write that the A-C linker is absent from the DC and therefore they will look at the proteins that localize at the luminal scaffold. Would it be possible to have a reference on the absence of the A-C linker in the spermatozoon? In a recent article published in BioRxiv by one of the authors of this paper, they identify very well densities that correspond to the A-C linker:

<https://www.biorxiv.org/content/10.1101/2020.11.18.388975v1.full.pdf>

- In a previous study published by the corresponding author, they identified that POC5, CENT1, and POC1B localizes into two-rod structures at the level of the DC in spermatozoon. Other more recent studies have shown that these proteins are part of a complex with FAM161A and WDR90 and form the inner scaffold in the centriole. In the manuscript, the authors refer to a “luminal scaffold” and not an “inner scaffold”. For the sake of consistency in the field, the authors should use the term “inner scaffold” because otherwise, it is quite confusing.

- Using SR, the authors localize FAM161A as part of the complex in the sperm cell, confirming the putative conservation of the inner scaffold. However they used only confocal microscopy to localize WDR90, would it be possible to use SR as well to precisely map this protein at the level of the luminal rod?

- The authors have already shown is that DCs have doublet microtubules that do not seem to be connected to each other in a previous publication. From the cryoET data, would it be possible to know the distance between each doublet microtubule, and do they see the absence of the inner scaffold structure ?

- The authors then identify that the inner scaffold proteins are located asymmetrically, which is new information compared to their previous publication which showed that POC5 and CENT1 had a V-shape. However, they had identified that this V-shape corresponds to only 40-50% of the observed forms. For example, POC5 is only visible at 20-40% in V-shape. Are the observations here different or have data been excluded?

- It is not clear where these inner scaffold proteins are located in the DC. The authors use cryoET to show the location of BAR structures but do not show the position of the doublets. It would be nice to make a summary diagram in Figure 1 that uses the dimensions and positions of the proteins in a model made from the positions of the doublets, bars, and other structures observed in cryoET. This would help to understand where each protein is located in relation to the observed structures.

- - In Figure 1 and in the supplement, the authors use a beautiful 3D rendering to show the overall structure coming from the cryoET. Would it be possible to show this image and the movie without the structure around it in white so that the doublet microtubules are more visible? This will allow the movement of the doublets to be clearly shown in the video.

- In figure 2, the authors use inner scaffold proteins as a proxy to analyze DC motion during flagella beating. The data are very nice and convincing. In a supplementary movie, the authors use a succession of images to reconstruct the DC motion. Would it be possible to have a zoom on the DC to see better the movement?

- In figure 3, the authors show that DC is not the only structure to be deformed by flagella beating but also the BAR structures and the densities which connect the axoneme to the sperm head. These results make sense and the data in cryoET are beautiful. As indicated above, several versions of the supplementary movie would be needed to isolate each element and thus better see their movements.

- this article is quite short with a lot of additional data which are for some difficult to understand because may or may not be detailed in the text. It would be necessary to give more details about the supplementary data. I also think Extended data 9 is beautiful and could be in the main figures because it shows very well the movement of the striated columns and the proximal centriole. Other figures could be in the main. Note also that I could not find which staining is used in Extended Data Fig. 3.

- On a more general level, it is difficult to really understand if these movements are only a consequence of the flagellar movement or if it "modulates" the flagellar movement as proposed by the authors. Likewise, the term "mechanotransducer" in the abstract, even if it is a hypothesis, is too strong and should only be proposed in the discussion. Same for "morphological computer".

Reviewer #2 (Remarks to the Author):

The manuscript by Khanal et al. presents imaging data from mammalian sperm, primarily bovine, that convincingly demonstrate the asymmetric and dynamic structure of the neck region and a correlation between asymmetry of structure and asymmetry of motions during swimming. Taken together, the statistical analyses of the imaging results provide support for a "sliding microtubules" model for sperm motility.

The imaging modalities used include STORM and deconvolution ("HyVolution") superresolution fluorescence, cryo-electron tomography, phase-contrast optical microscopy and confocal fluorescence. On the whole the imaging quality and analysis are good, but an exception is the "3D" STORM imaging, which seems to contain primarily 2D information, with very little information provided by the putative Z localizations.

A few minor changes could improve the manuscript:

1) There should be a more detailed explanation (1 or 2 sentences) describing the structure of canonical centrioles, including microtubule arrangement. That way it would be easier for non-expert readers to understand why the DC is atypical.

2) There should be an explanation somewhere on what the actual basal sliding hypothesis is, and then how the results presented confirm this. Perhaps an animation in Supplemental Material would help.

3) Several minor points of wording or technical concerns:

a. "In most eukaryotes, the PC is a canonical cylindrical centriole 9,14. However, in mammals, the DC, at the base of the flagellum, is structurally atypical 15. Since, similar to canonical centrioles, the atypical centriole functions post-fertilization, the reason for its atypical structure remains unknown 16-18." Please reword. It isn't clear from the wording that the PC is canonical in the sperm you are

studying. What is the function of the centrioles post-fertilization? Maybe re-word to say “The most notable function of the centrioles is to ... post-fertilization, however the reason for the atypical structure of the DC in mammalian sperm remains unknown.”

b. “The PC and DC of mammalian sperm are embedded in a specialized mass of atypical pericentriolar material: the segmented columns (SCs) and the capitulum.”

c. “In the spermatozoon, the DC consists of doublets instead of triplets and lacks the A-C linkers, while the doublets are splayed apart, suggesting increased DC compliance.” Reference? Reference 15 does not prove this, though it is in their diagrams. Also, reference 15 shows 3 rods with POC5 staining using STORM. Can the authors address this?

d. “-This asymmetry agrees with the axoneme’s structural and functional asymmetry, where the left four axonemal microtubules work against the right three axonemal microtubules, generating a stronger left torque 24,26.” This suggests that there are only 7 microtubule doublets in the DC? Why are there not 9? And a reference for that as well.

e. In Fig. 1, the color-code for Z depths seems to be of little value. It is not clear that the values generated by the Elements software are accurate, as no calibration is reported, and almost all signal seems to come from the upper half of the 1178 nm range (no red or orange pixels, only a few yellow ones).

f. The definition of “Left” and “Right” given on page 4 of the Supplemental Material should be reiterated or referred to here.

g. In Fig. 1f, the capitulum and basal plate are not labeled. In the cartoon in the lower portion of Fig. 1a, the PC size seems much smaller than its relative size in the tomogram as seen in 1f.

h. Fig. 1g: the labels “1” and “2” and “3” are not identified.

i. Typo in legend for 1b-e: “statistical” for “statistics.”

j. Typo page 4 of Supplemental: “laying” for “lying.”

k. Supplemental Fig. 1a-b legend: “Co-localization” is only partial, even at the resolution shown.

Reviewer #1:

In this study, Khanal and colleagues use super-resolution microscopy (SRM) and volta phase plate cryo-tomography (cryoET) to demonstrate that the region between the head and the tail of the sperm cell (head-tail junction) is deformed upon tail beating and head kinking. It is a very beautiful analysis that combines state-of-the-art imaging techniques and demonstrates for the first time that distal centriole (DC) deforms under the effect of flagellar beating and that the surrounding protein complexes also undergo deformation. These observations provide a better understanding of what happens at the nanoscale during the flagellar beating. I think that this article is of high importance, nevertheless, I think that some points need to be clarified before publication:

We appreciate the reviewer's kind comments and support for this paper.

- Initially, the authors localize several proteins at the DC level. They write that the A-C linker is absent from the DC and therefore they will look at the proteins that localize at the luminal scaffold. Would it be possible to have a reference on the absence of the A-C linker in the spermatozoon? In a recent article published in BioRxiv by one of the authors of this paper, they identify very well densities that correspond to the A-C linker: <https://www.biorxiv.org/content/10.1101/2020.11.18.388975v1.full.pdf>

We appreciate the reviewer's point that the subtomogram averages we reported (now published, EMBO J 2021 40:e107410) for the proximal centriole (PC) shows density for the A-C linker attached to the C-tubule of triplets (yellow), as well as some density for the putative A-link associated with protofilaments A8/A9 (olive). In contrast, the DC consists of doublets and thus lacks both the C-tubule and the associated portion of the A-C linker. The A-tubule of the DC does have additional density in roughly the same position as the A-link (i.e; protofilaments 8 and 9, sky blue), but because neighboring triplets do not have a C-tubule, these would not form the A-C linker. We have added the reference and modified the text to clarify our point and the reviewer's comment.

- In a previous study published by the corresponding author, they identified that POC5, CENT1, and POC1B localizes into two-rod structures at the level of the DC in spermatozoon. Other more recent studies have shown that these proteins are part of a complex with FAM161A and WDR90 and form the inner scaffold in the centriole. In the manuscript, the authors refer to a "luminal scaffold" and not an "inner scaffold". For the sake of consistency in the field, the authors should use the term "inner scaffold" because otherwise, it is quite confusing.

We appreciate the reviewer's concern for consistency and made the suggested change.

- Using SR, the authors localize FAM161A as part of the complex in the sperm cell, confirming the putative conservation of the inner scaffold. However they used only confocal microscopy to localize WDR90, would it be possible to use SR as well to precisely map this protein at the level of the luminal rod?

As suggested, we have tried to perform STORM in bovine and human sperm, but the antibody signal against the WDR90 is weak in confocal and is too weak for STORM imaging. We added this point in the method. A similar weak signal was observed in a recent paper¹.

- The authors have already shown is that DCs have doublet microtubules that do not seem to be connected to each other in a previous publication. From the cryoET data, would it be possible to know the distance between each doublet microtubule, and do they see the absence of the inner scaffold structure?

Thank you for the suggestion.

Unfortunately quantifying the doublet microtubules of the DC is not straightforward for several reasons:

- (1) The DC consists of doublet microtubules splayed out asymmetrically around a pair of singlets. The doublets do not all start at the same location (even within the same cell; visible in some of our segmentations), so it would be difficult to define a point of reference that would allow robust comparisons across tomograms.**
- (2) The bull sperm neck is thick because mitochondria extend quite far proximally. Because of sample thickness and the fact that we imaged with a 200 kV microscope in this study, even with the Volta phase plate, the DC doublets are not all equally well-resolved across tomograms, and sometimes even within the same tomogram.**

Because of these limitations, we were very conservative and chose not to make any strong claims about the doublets of the DC beyond the fact that they are splayed out around the central singlets. Hence, we used relative movement of the rod proteins as a proxy to analyze sliding in the DC, and supplemented this by using cryoET to analyze movement in the connecting piece/striated columns.

With the current cryoET data, we cannot visualize the inner scaffold and cannot comment on whether it is entirely absent. This is, of course, an excellent question that merits further investigation. However, the resolution needed to address this likely entails cryo-focused ion beam milling, which is beyond the scope of the current study.

- The authors then identify that the inner scaffold proteins are located asymmetrically, which is new information compared to their previous publication which showed that POC5 and CENT1 had a V-shape. However, they had identified that this V-shape corresponds to only 40-50% of the observed forms. For example, POC5 is only visible at 20-40% in V-shape. Are the observations here different or have data been excluded?

We appreciate the reviewer's discussion on the frequency of V-shape morphology in the DC. In the current paper, we observe V-shape (or V-shape with sliding) in 82% (total n=140) of bovine sperm with FAM161A staining. This higher frequency is due to several differences between the 2018 paper and the current work regarding V-shape frequency relative to other shapes.

1) The POC5 and CENT1 V-shape in the 2018 paper were observed in humans. The V-shape described in the current paper is observed in bovine. So, we cannot compare the two.

2) Human sperm are much more heterogeneous than bovine sperm explaining the more diverse rod shapes in humans.

3) The current study observed that the rods move relative to each other in coordination with the tail bending can explain some of the non-v shapes observed in humans.

We thank the referee for raising this point and we added a new supplementary figure that summarizes this data for clarity - Supplementary Fig. 1.

- It is not clear where these inner scaffold proteins are located in the DC. The authors use cryoET to show the location of BAR structures but do not show the position of the doublets. It would be nice to make a summary diagram in Figure 1 that uses the dimensions and positions of the proteins in a model made from the positions of the doublets, bars, and other structures observed in cryoET. This would help to understand where each protein is located in relation to the observed structures.

Thank you for the suggestion. We added a model as Fig. 2e depicting the relative position of doublet microtubules, rods, and bars that we observed based on STORM and cryo-ET imaging.

We see an elongated density associated with the inner surface of the lateral DC doublets. The location of this density corresponds to what would be expected from our STORM measurements. We added this point as a new Supplementary Fig. 2e using the cells shown in Fig 2f and a new cell.

- In Figure 1 and in the supplement, the authors use a beautiful 3D rendering to show the overall structure coming from the cryoET. Would it be possible to show this image and the movie without the structure around it in white so that the doublet microtubules are more visible? This will allow the movement of the doublets to be clearly shown in the video.

Thank you for the suggestion. We added a new Supplementary video 2b and a new 3D rendering in Fig. 2f. Note that instead of completely removing the segmented columns, we made them fully transparent but kept an outline to make it easier to interpret. Also, note that we cannot make strong conclusions about the cryoET data's doublets at this point (see points above). We stress that what readers should pay attention to in the cryoET gif are: 1) movement of the segmented columns, (2) movement of the PC relative to the DC, (3) movement (or rather lack thereof) of the bars. Furthermore, we note that the gif is for illustration purposes only as it comes from three different cells and does not capture one cell's movement. We have further clarified these in the descriptions of the videos uploaded.

- In figure 2, the authors use inner scaffold proteins as a proxy to analyze DC motion during flagella beating. The data are very nice and convincing. In a supplementary movie, the authors use a succession of images to reconstruct the DC motion. Would it be possible to have a zoom on the DC to see better the movement?

Thank you for the suggestion. We added Supplementary video 1 with DC zoomed in for rod movement.

- In figure 3, the authors show that DC is not the only structure to be deformed by flagella beating but also the BAR structures and the densities which connect the axoneme to the sperm head. These results make sense and the data in cryoET are beautiful. As indicated above, several versions of the supplementary movie would be needed to isolate each element and thus better see their movements.

Thank you for the suggestion. As described above, we added a new Supplementary video 2b.

- this article is quite short with a lot of additional data which are for some difficult to understand because may or may not be detailed in the text. It would be necessary to give more details about the supplementary data. I also think Extended data 9 is beautiful and could be in the main figures because it shows very well the movement of the striated columns and the proximal centriole. Other figures could be in the main. Note also that I could not find which staining is used in Extended Data Fig. 3.

Thank you very much for the suggestion. We moved Extended Data Fig. 1 from our previous version manuscript to main Fig. 1, Extended Data Fig. 9 to main Fig. 4. We also moved some of the data from Extended Data Fig. 11 to main Fig. 5.

We apologize for forgetting to indicate that FAM161A was the staining marker in Extended Data Fig. 3—now Supplementary Fig 3 in the new version. This issue is now corrected.

- On a more general level, it is difficult to really understand if these movements are only a consequence of the flagellar movement or if it "modulates" the flagellar movement as proposed by the authors. Likewise, the term

"mechanotransducer" in the abstract, even if it is a hypothesis, is too strong and should only be proposed in the discussion. Same for "morphological computer".

We thank the referee for this important suggestion. We have now removed the terms "mechanotransducer" and "morphological computer" from the abstract and relocate them to the discussion. We also add the point of the reviewer to clarify that this is a hypothesis.

Reviewer #2

The manuscript by Khanal et al. presents imaging data from mammalian sperm, primarily bovine, that convincingly demonstrate the asymmetric and dynamic structure of the neck region and a correlation between asymmetry of structure and asymmetry of motions during swimming. Taken together, the statistical analyses of the imaging results provide support for a "sliding microtubules" model for sperm motility.

The imaging modalities used include STORM and deconvolution ("HyVolution") superresolution fluorescence, cryo-electron tomography, phase-contrast optical microscopy and confocal fluorescence. On the whole the imaging quality and analysis are good, but an exception is the "3D" STORM imaging, which seems to contain primarily 2D information, with very little information provided by the putative Z localizations.

We appreciate the reviewer's kind comments and support for this paper and the concern for Z localizations related to the STORM imaging. We did not emphasize the Z localizations because the rods and microtubules of the DC are organized in a narrow plane, as now shown in a new Supplementary video 4.

In light of the reviewer's comment, we now added Supplementary Videos 4a, 4b, 4c, and 4d with 3D rotation of the cell stained for rod protein and microtubules and Supplementary Fig. 1e showing rods and microtubules at different angles while rotating in 3D. Please refer to detailed info on Z localizations in point e below.

A few minor changes could improve the manuscript:

1) There should be a more detailed explanation (1 or 2 sentences) describing the structure of canonical centrioles, including microtubule arrangement. That way it would be easier for non-expert readers to understand why the DC is atypical.

We thank the referee for this suggestion. We have added an explanation describing the structure of canonical centrioles.

2) There should be an explanation somewhere on what the actual basal sliding hypothesis is, and then how the results presented confirm this. Perhaps an animation in Supplemental Material would help.

As suggested, we have added an explanation of the basal sliding hypothesis in the introduction and our supporting observations (Sfig. 6) (Svideo 2b) (Svideo 4a-d) showing the sliding of the whole neck structure.

3) Several minor points of wording or technical concerns:

a. "In most eukaryotes, the PC is a canonical cylindrical centriole 9,14. However, in mammals, the DC, at the base of the flagellum, is structurally atypical 15. Since, similar to canonical centrioles, the atypical centriole functions post-fertilization, the reason for its atypical structure remains unknown 16-18." Please reword. It isn't clear from the wording that the PC is canonical in the sperm you are studying. What is the function of the centrioles post-fertilization? Maybe re-word to say, "The most notable function of the centrioles is to ... post-fertilization, however the reason for the atypical structure of the DC in mammalian sperm remains unknown."

We thank the reviewer for these suggestions. We added that in the sperm of humans, bovines, and rabbits, the PC is more similar to the canonical cylindrical centriole. We also added that the most

notable function of the sperm centrioles is after fertilization and that they recruit PCM and form a centrosome that emanates a large microtubule aster that helps bring together the pronuclei.

b. “The PC and DC of mammalian sperm are embedded in a specialized mass of atypical pericentriolar material: the segmented columns (SCs) and the capitulum.”

We added a reference (Fawcett, 1969) to this statement.

c. “In the spermatozoon, the DC consists of doublets instead of triplets and lacks the A-C linkers, while the doublets are splayed apart, suggesting increased DC compliance.” Reference? Reference 15 does not prove this, though it is in their diagrams. Also, reference 15 shows 3 rods with POC5 staining using STORM. Can the authors address this?

We have reworded the section on the A-C linkers and added a new reference. We now state that the DC consists of splayed apart doublets instead of triplets; the A tubule has A-C-like densities that are not connected to the C tubule ² (See reply to reviewer 1 above).

We see two rods in 87% of the sperm cells and three rods in 13% of the cells stained for rod proteins (FAM161A, POC1B, or POC5) (see Supplementary Fig 1d). This data suggest the majority of sperm cells have two rods as we described in our model. In 2018 paper we reported only cells with three rods. This difference may be because 2018 paper looked on few cells but in this new manuscript we looked on over 200 sperm cells.

d. “-This asymmetry agrees with the axoneme's structural and functional asymmetry, where the left four axonemal microtubules work against the right three axonemal microtubules, generating a stronger left torque 24,26.” This suggests that there are only 7 microtubule doublets in the DC? Why are there not 9? And a reference for that as well.

We thank the reviewer for these comments and changed the text to clarify that there are nine microtubule doublets in the axoneme. We now clarify and state that this asymmetry agrees with the axoneme's structural and functional asymmetry, where the left four axonemal microtubules (doublets 4, 5, 6, and 7) work against the right three axonemal microtubules (doublets 1, 2, and 9) and the two stationary axonemal microtubules (doublets 3 and 8), generating a stronger left torque. References are cited for this axonemal organization.

e. In Fig. 1, the color-code for Z depths seems to be of little value. It is not clear that the values generated by the Elements software are accurate, as no calibration is reported, and almost all signal seems to come from the upper half of the 1178 nm range (no red or orange pixels, only a few yellow ones).

We thank the reviewer for these comments. We added that calibration is done using florescent beads and stored as a file used for each specific objective and buffer condition according to manufacturer instruction. In our study, STORM imaging was done consistently with the same 100X objective and buffer composition.

Also, we added here a picture from the sperm head. Please note that the color range covers 700 nm, consistent with the bovine nucleus thickness.

f. The definition of “Left” and “Right” given on page 4 of the Supplemental Material should be reiterated or referred to here.

Many thanks for this suggestion. We have now reiterated the definition of “Left” and “Right. We state that sperm cells images were rotated to view the neck straight, were oriented to maintain the PC tip pointing to the right side, and bigger rod on the left side of the neck midline. This consistent reference was used for defining the left-right rod and microtubules.

g. In Fig. 1f, the capitulum and basal plate are not labeled. In the cartoon in the lower portion of Fig. 1a, the PC size seems much smaller than its relative size in the tomogram as seen in 1f.

Thank you for the comment. We corrected the size of the PC, and we also labeled Ca and Bp in previous version manuscript Fig. 1f, current version manuscript Fig. 2f.

h. Fig. 1g: the labels “1” and “2” and “3” are not identified.

We apologize for missing these identifications. We now added that the labels “1” and “2” and “3” mark distinct Bar’s plates

i. Typo in legend for 1b-e: “statistical” for “statistics.”

Now corrected.

j. Typo page 4 of Supplemental: “laying” for “lying.”

Now corrected.

k. Supplemental Fig. 1a-b legend: “Co-localization” is only partial, even at the resolution shown.

We now pointed that the proteins partially colocalize.

References to the P by P

- 1 Steib, E. *et al.* WDR90 is a centriolar microtubule wall protein important for centriole architecture integrity. *eLife* **9**, e57205, doi:10.7554/eLife.57205 (2020).
- 2 Leung, M. R. *et al.* The multi-scale architecture of mammalian sperm flagella and implications for ciliary motility. *Embo j*, e107410, doi:10.15252/embj.2020107410 (2021).

Reviewer #1 (Remarks to the Author):

The answers and modifications/corrections made by de Khanal and colleagues are satisfactory and the additional data added greatly improves the understanding and visualization of the movements they describe in their article. Therefore, I highly recommend the paper for publication.

Reviewer #2 (Remarks to the Author):

The additional information in the manuscript and the supplemental files have adequately addressed all the concerns raised in the previous round of review.